# Cryo-electron microscopy of the f1 filamentous phage reveals insights into viral infection and assembly

Rebecca Conners [1,2,5], Rayén Ignacia León-Quezada [3,4,5],
Mathew McLaren[1,2,5], Nicholas J. Bennett [3], Bertram Daum [1,2],
Jasna Rakonjac [3,4,6] ✉ & Vicki A. M. Gold [1,2,6] ✉

Phages are viruses that infect bacteria and dominate every ecosystem on our planet. As well as impacting microbial ecology, physiology and evolution, phages are exploited as tools in molecular biology and biotechnology. This is particularly true for the Ff (f1, fd or M13) phages, which represent a widely distributed group of filamentous viruses. Over nearly five decades, Ffs have seen an extraordinary range of applications, yet the complete structure of the phage capsid and consequently the mechanisms of infection and assembly remain largely mysterious. In this work, we use cryo-electron microscopy and a highly efficient system for production of short Ff-derived nanorods to determine a structure of a filamentous virus including the tips. We show that structure combined with mutagenesis can identify phage domains that are important in bacterial attack and for release of new progeny, allowing new models to be proposed for the phage lifecycle.

Filamentous phages are widely distributed viruses infecting all bacterial genera and some archaea[1]. A number of filamentous phages infecting Gram-negative bacteria have been implicated in virulence, for example horizontal gene transfer of cholera toxin in *Vibrio cholerae*[2], or biofilm formation in *Pseudomonas aeruginosa*[3]. The phages are approximately one micron long and 6–7 nm wide–visualised as hair-like filaments by electron microscopy[4]. A notable feature of the filamentous phage lifecycle is that they replicate and egress without killing their bacterial host. Most extensively studied in this group are the F-specific filamentous phages (Ffs) that include viruses f1, fd and M13. Ffs infect cells by binding to the F-pilus of *E. coli*[5]. The phages are 98.5% identical in their DNA sequence and are used interchangeably. Ffs are also strikingly simple–their single-stranded DNA genome encodes just 11 proteins, of which 5 form the phage capsid. This simplicity, plus their high stability, has facilitated the use of Ffs in modern biotechnology–for example in phage display, as a vaccine carrier, in tissue engineering, as a high-powered lithium battery, and in phage therapy[6–9]. Despite their extensive downstream applications, the structure of the complete phage capsid assembly remains a mystery[10].

pVIII is the major capsid protein that forms the filamentous body of the phage. The tips of the filament are formed of the minor capsid proteins which occur in pairs. pIII and pVI form the leading end of the infecting phage, which interacts with the host receptors to initiate entry, whereas pVII and pIX form the trailing end (Fig. 1a).

In order for the phage to inject its DNA into the bacterial cytoplasm, it must navigate both the *E. coli* outer and cytoplasmic membranes. Adsorption of the phage to the F-pilus is initiated by the minor capsid protein pIII, which has been divided into 3 domains: N1, N2 and C (Fig. 1b). Specifically, the N2 domain binds to the F-pilus tip[13] (Fig. 1a), which has two important effects: (1) the phage is pulled towards the bacterial surface on F-pilus retraction; and (2) a conformational change occurs in pIII[14,15]. This frees the N1 domain, enabling it to bind to the periplasmic domain of the secondary receptor TolA, which is part of the TolQRA complex embedded in the cytoplasmic membrane[16]

[1]Living Systems Institute, University of Exeter, Stocker Road, Exeter EX4 4QD, UK. [2]Faculty of Health and Life Sciences, University of Exeter, Exeter EX4 4QD, UK. [3]School of Natural Sciences, Massey University, Palmerston North, New Zealand. [4]Nanophage Technologies, Palmerston North, New Zealand. [5]These authors contributed equally: Rebecca Conners, Rayén Ignacia León-Quezada, Mathew McLaren. [6]These authors jointly supervised this work: Jasna Rakonjac, Vicki A. M. Gold. ✉e-mail: j.rakonjac@massey.ac.nz; v.a.m.gold@exeter.ac.uk

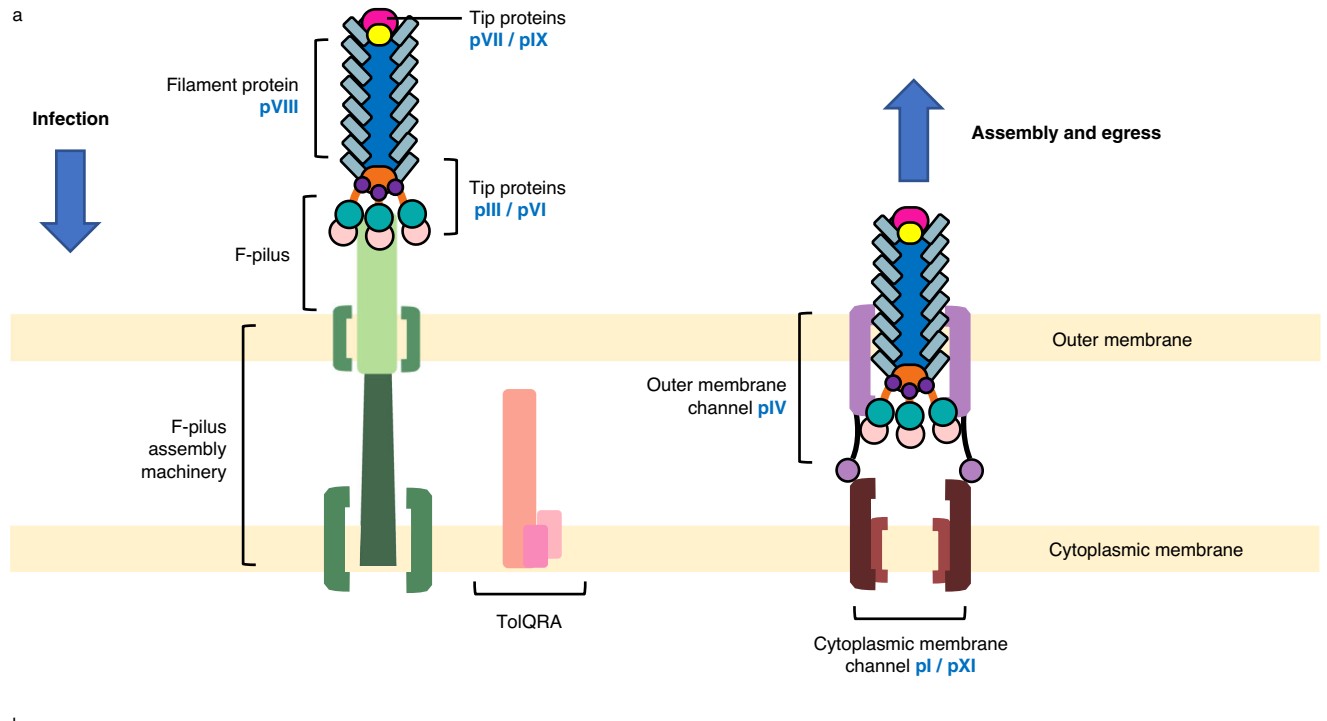

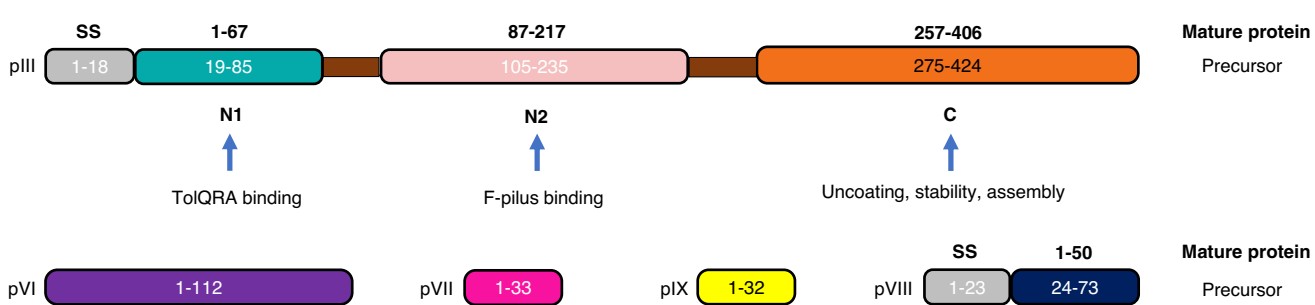

**Fig. 1 | Schematic of phage infection, egress and the proteins that comprise the phage capsid. a** f1 is comprised of the major capsid protein pVIII (light blue rectangles), which forms the filamentous body of the phage. The minor capsid proteins pIII (C domain orange oval, N1 domain teal circles and N2 domain light pink circles) and pVI (purple circles) form one tip, with pVII (pink oval) and pIX (yellow oval) forming the other. For clarity, only 3 pIII/pVI proteins and one each of pVII/pIX are shown, instead of 5 of each. pIII binds to the F-pilus (light green), which on retraction, ultimately allows the phage to reach the TolQRA complex anchored in the cytoplasmic membrane (shades of pink). After DNA injection and genome replication, phage proteins are expressed and assembled. pI/pXI (shades of brown)

in the cytoplasmic membrane align with the outer membrane secretin pIV (mauve), potentially via the N0 domain of pIV (mauve circles), connected to the pIV barrel via flexible linkers[19]. Phage egress is dependent on the proton-motive force and ATP hydrolysis by pI. **b** Schematic to show the f1 capsid proteins and their domains. The numbering for the pIII and pVIII precursor proteins is shown, alongside the numbering for the mature proteins (bold). The domains of pIII are coloured as follows: N1 (teal), N2 (light pink), C (orange) and glycine-rich linkers (brown). We have used the numbering for the mature protein throughout the manuscript. SS signal sequence.

(Fig. 1a). The mechanism by which the phage is able to cross the two membranes is unknown.

The C domain of pIII is involved in virion uncoating and DNA entry into the host cell cytoplasm[17] (Fig. 1b), where the major capsid protein pVIII is stripped off and integrates into the cytoplasmic membrane[18]. The host transcription and translation machinery replicates the Ff genome, and phage-encoded proteins are synthesised.

Newly synthesised Ff particles exit the cell through a phage-encoded transmembrane egress machinery, comprised of proteins pI and pXI in the cytoplasmic membrane, and pIV in the outer membrane (Fig. 1a). We recently determined the structure of pIV by cryo-electron microscopy (cryoEM), revealing how the gates in the channel would need to open to allow phage egress[19].

All phage capsid proteins are synthesised as integral membrane proteins[20]. The DNA packaging signal is a hairpin, which interacts with the C-terminal residues of the tip proteins pVII and pIX to enable DNA

to be incorporated into new virions[21]. The major coat protein pVIII is then added, until the entire genome is covered. pIII and pVI are subsequently added as a terminating cap and the phage is released[22,23]. The phage is now in reverse orientation, with the tip proteins pVII and pIX forming the leading end of the egressing filament, and the tip proteins pIII and pVI forming the trailing end[11] (Fig. 1a).

From a structural perspective, the major capsid protein pVIII is the most well-studied to date, with experiments employing fibre diffraction[24–26], NMR[27], and cryoEM[28]. The four minor capsid proteins have not however been amenable to structural characterisation, with the exception of N-terminal fragments of pIII[14,34]. Therefore, how the capsid proteins interact with each other to form the assembled phage remains elusive. Currently, cryoEM is the only technique that could be employed to determine a high-resolution structure of an entire filamentous virus. However, determination of protein structures by cryoEM requires averaging thousands of copies of the protein of

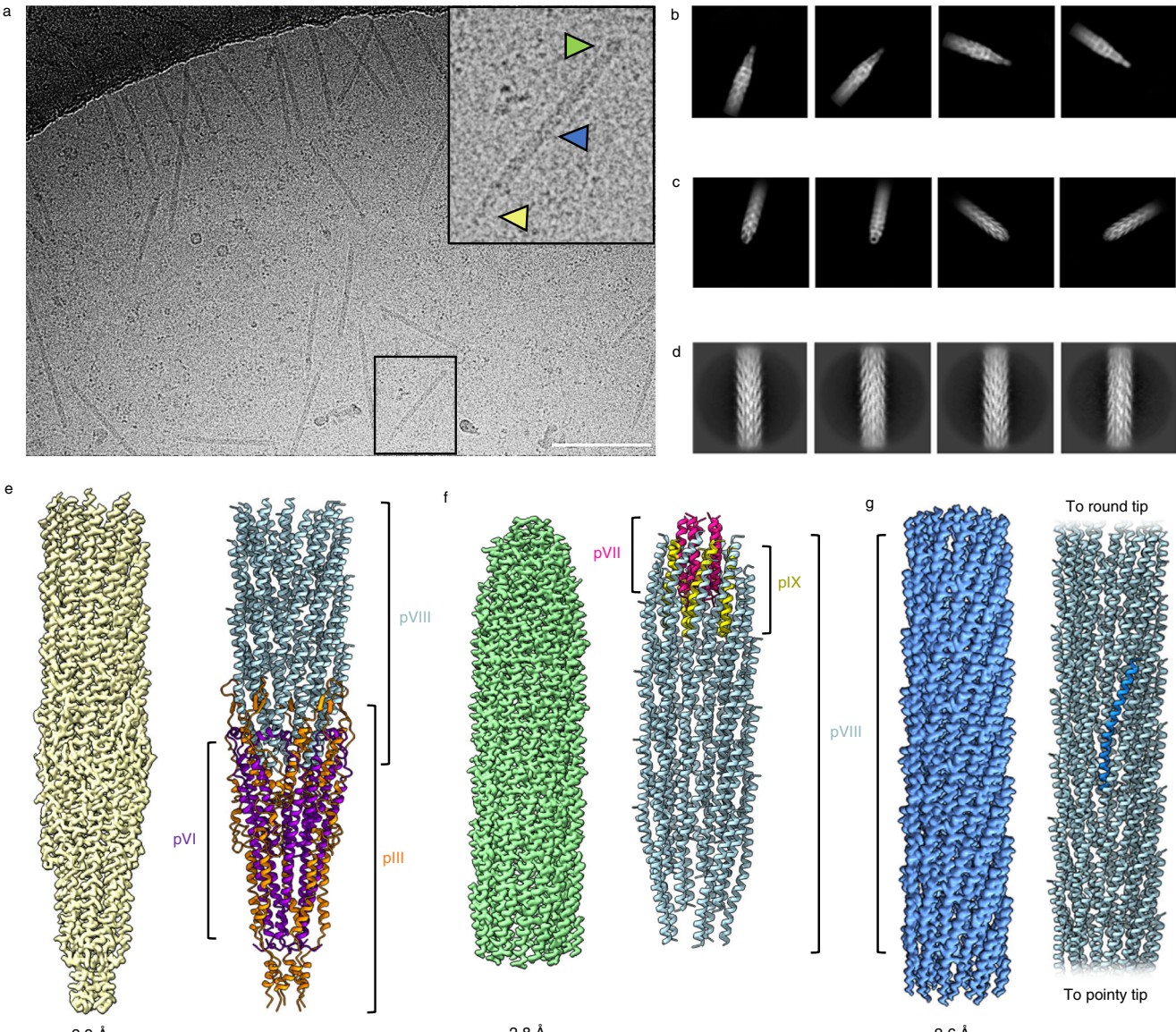

**Fig. 2 | Determination of phage capsid protein structures. a** Electron micrograph of nanorods with an enlarged view (boxed) showing the pointy tip (yellow arrowhead), round tip (green arrowhead) and central filamentous section (blue arrowhead). Scale bar, 100 nm. The image is representative of 25,065 micrographs collected from a single sample. **b–d** Selected 2D class averages from cryoSPARC, showing **b** four pointy classes, **c** four round classes and **d** four classes of the central filamentous section. **e** Final 3D reconstruction of the pointy tip (yellow) and corresponding ribbon diagram with pIII shown in orange, pVI in purple and pVIII in light blue. **f** Final 3D reconstruction of the round tip (green) and corresponding ribbon diagram with pVII shown in pink, pIX in yellow and pVIII in light blue. **g** Final 3D reconstruction of the central filamentous section (bright blue) and corresponding ribbon diagram with protein pVIII shown in light blue. A single subunit is highlighted in bright blue.

interest. Due to the length of Ffs (~1 μm)[4], the tips are usually present at low abundance in high-magnification images, which makes structural determination challenging. However, the length of Ffs can be modulated by changing the size of the packaged DNA[29].

In this work, we develop a phage-free plasmid-based system that enables short versions of Ff to be produced based on f1 phage, referred to as nanorods. The technology enables us to determine the high-resolution structure of an assembled filamentous phage using cryoEM, allowing biochemical data to be reconciled and models proposed for infection, assembly and egress.

## Results

### Generation of f1-derived nanorods

Nanorods were produced using a two-plasmid system (Supplementary Fig. 1a, b). A nanorod template plasmid was used to generate circular

single-stranded DNA 529 nucleotides long, alongside a helper plasmid encoding all f1 phage proteins. On transformation of both plasmids into bacteria, 529-nt circular ssDNA was produced and subsequently packaged into short phage-like nanorod particles by the f1 proteins expressed from the helper plasmid. pVIII contained a Y21M replacement shown previously to result in a more stable conformation of the major coat protein[26]. Nanorods egressed from bacterial cells were purified from the medium using CsCl density gradient centrifugation followed by anion exchange chromatography (Supplementary Fig. 2a–c).

### Structure determination of the f1-derived nanorod capsid

Purified nanorods were vitrified by plunge-freezing, and data were collected by cryoEM (Supplementary Table 1). In micrographs, nanorods were observed as structures of ~80 nm in length (Fig. 2a, Supplementary Fig. 2d, e), with two distinctive ends: one pointy and

one round (Fig. 2a). Automated particle picking was performed using Warp[30] for the tips and cryoSPARC[31] for the filament, and both datasets were subsequently processed using cryoSPARC. After 2D classification of the tips (Fig. 2b, c, Supplementary Fig. 3a), particles could clearly be seen to belong to either the pointy or round classes and were sorted accordingly. Additional 2D classification of the filament (Fig. 2d, Supplementary Fig. 3b) allowed three different cryoEM maps of the phage to be obtained from a single dataset. All maps had 5-fold symmetry applied in the final reconstruction. The map of the central, filamentous region of the capsid had additional helical symmetry applied. The helical parameters were determined using cryoSPARC and were refined to a helical twist of 37.44° and a helical rise of 16.6 Å. The final 3D reconstructions produced maps with resolutions of 2.9 Å, 2.8 Å and 2.6 Å for the pointy, round and central filamentous structures, respectively (Supplementary Fig. 4), into which atomic models were built (Fig. 2e–g, Supplementary movies 1–3). Further details of structure determination are given in Methods.

## Structure of the pIII/pVI pointy tip

The pointy tip is comprised of proteins pIII and pVI, present in 5 copies each (Figs. 2e and 3). pIII is the largest and most structurally complex capsid protein. It is produced with a signal sequence which is cleaved to leave the mature protein of 406 amino acids[32]. Its three domains (N1, N2 and C; of 67, 131 and 150 residues, respectively) are separated by two flexible glycine-rich linkers (Fig. 1b). In our map, the N1-N2 domains and the glycine linkers are not visible, but density is seen for residues 257–404 of the C domain (accounting for the entire C domain minus the final two residues). The pIII C domain is comprised of several α-helices of differing lengths (Fig. 3a). Two β-strands form a hairpin loop which extends over the exterior of the main filamentous part of the virion; reminiscent of sepals protecting a flower bud (box 1 within Fig. 3a). We observed two cysteine residues (354 and 371) at the start and finish of the β-hairpin loop, in close enough proximity to form an intrachain disulphide bond. A disulphide bond was not visualised in the map, likely due to sensitivity of disulphide bonds to electron radiation[33]. A disulphide bond in this position would pin the β-hairpin loop together and further reinforce the structural motif. pIII is predicted to contain three additional disulphide bonds; two in the N1 domain (between residues 7 and 36, and 46 and 53) and one in the N2 domain (between residues 188 and 201)[34]. The C-terminus of pIII is buried in the centre of the tip, where the 5 symmetry copies come together to form a distinct stricture in the virion lumen (box 2 within Fig. 3a, Supplementary Fig. 5a). The lumen measures 8.4 Å across at this point, compared to 20 Å across at its widest point. We observe hydrogen bonds between neighbouring pIII molecules, pinning the tip together (Supplementary Fig. 5b). Two additional interchain hydrogen bonds are made between neighbouring pIII subunits; both are situated near rings of methionine residues at the terminus of the pointy tip (Supplementary Fig. 5c).

pVI is a 112-residue, mostly hydrophobic protein which lacks a signal sequence (Fig. 1b). Density was observed for the entire pVI protein, which is composed of 3 α-helices arranged in a U shape (Fig. 3a). The N-terminus of the protein forms the longest α-helix of 54 residues in length, which turns into a short 10 residue α-helix, and then into the final C-terminal α-helix of 32 residues. The C-terminus of pVI is buried in the centre of the pointy tip. Each pVI chain forms seven hydrogen bonds with neighbouring pVI chains; four with one neighbouring chain and three with the other. These bonds are mostly concentrated in the area at the centre of the virion near the pVI C-termini (Supplementary Fig. 5d).

pIII and pVI can be seen to interact extensively with each other, within a network of closely-packed helices (Fig. 3). Five copies of each of the proteins are intertwined and arranged symmetrically to form the pointy tip. pIII makes 12 hydrogen bonds and 1 salt bridge with its two neighbouring pVI molecules (Fig. 3b). The outer surface of pVI (which is

pIII-facing) is hydrophobic with two distinct rings of positive charge (Fig. 3c, Supplementary Figs. 6, 7). pIII forms a negatively charged scaffold around it, with hydrophobic interactions being shielded at the centre of the assembled phage (Fig. 3c, Supplementary Figs. 6, 7). The pointy tip is overall negatively charged and hydrophilic on the solvent-exposed outside.

At the pointy tip, there were three areas of density that were unaccounted for in the cryoEM map, lying within the central pore of the virion (Supplementary Fig. 8a). The first area, at the very tip of the structure, is surrounded by a ring of phenylalanine residues, a lower ring of methionine residues and a third ring of asparagine residues from the pIII protein (Supplementary Fig. 8b). The second, smaller area of density is adjacent to the first, with the asparagine ring defining one end and a second methionine ring the other end (Supplementary Fig. 8b). The third area is located closer to the main body of the filament, where 5 symmetrical curved tubes of density were observed lining a hydrophobic section of the lumen formed by the pVI protein (Supplementary Fig. 8c). These tube-like densities are all consistent with the size and shape of fatty acid molecules, which is supported by the knowledge that lipids are not uncommon in the capsids of many different types of virus[35]. Interestingly, lipids have also been observed tightly bound to a number of filamentous bacterial and archaeal mating pili[36]. The binding environment provided by the phage corroborates our observation, with hydrophobic residues lining most of the pocket, and hydrophilic residues available to stabilise the acid group of the fatty acid (Supplementary Fig. 8c). Fatty acids were modelled into the map according to their size, suggesting one molecule of octanoic acid (C8) at the tip, followed by one molecule of butanoic acid (C4) and then 5 symmetry-related molecules of dodecanoic acid (C12) in the third site (Supplementary Fig. 8b, c). Lipids are usually acquired in phages during their assembly and are thought to aid the infection process by diverse mechanisms[35].

## Structure of the pVII/pIX round tip

pVII and pIX are small hydrophobic proteins of only 33 and 32 amino acids, respectively (Fig. 1b). They both lack a signal sequence, form a single α-helix, and are present at the tip in 5 copies each (Fig. 4a). In comparison to the pointy tip, the round one has a much simpler composition. 5 copies of pVII form a helical bundle, packed together using mainly hydrophobic interactions (Fig. 4a, Supplementary Fig. 9). Hydrophobic residues line the sides of the helices that pack together, and the interior of the bundle is also predominantly hydrophobic. The terminus of the round end of the phage is comprised of the N-terminus of pVII, which is mostly negatively charged (Fig. 4a, Supplementary Fig. 10). In our map, density is missing for five residues as the far N-terminus of pVII, which would contribute to the negative charge and be of mixed hydrophobicity. The pVII helices have typical intrahelical hydrogen bonding, with two interchain hydrogen bonds between neighbouring subunits (Fig. 4b). All proteins are arranged with their positively charged C-termini oriented towards the DNA and the main body of the phage (Fig. 4a). This arrangement explains why display of peptide or protein at the round tip of Ff is possible only when linked to the N-termini of pVII or pIX[12].

## Structure of the pVIII filament

pVIII forms the main body of the filamentous phage. It is produced initially with a signal sequence that is cleaved to leave a single α-helix of 50 residues in length (Fig. 1b), with thousands of these helices packing together in a helical array to coat the length of the phage DNA genome (Fig. 2g). The α-helices are slightly curved; with their C-termini projecting into the phage lumen, resulting in an overall highly positive charge (Fig. 4c, Supplementary Figs. 7d, 10d). The N-terminus of pVIII is splayed outwards towards the exterior of the capsid. In our map, density is smeared for the four N-terminal residues, indicating that these are flexible. The missing sequence would be overall negatively

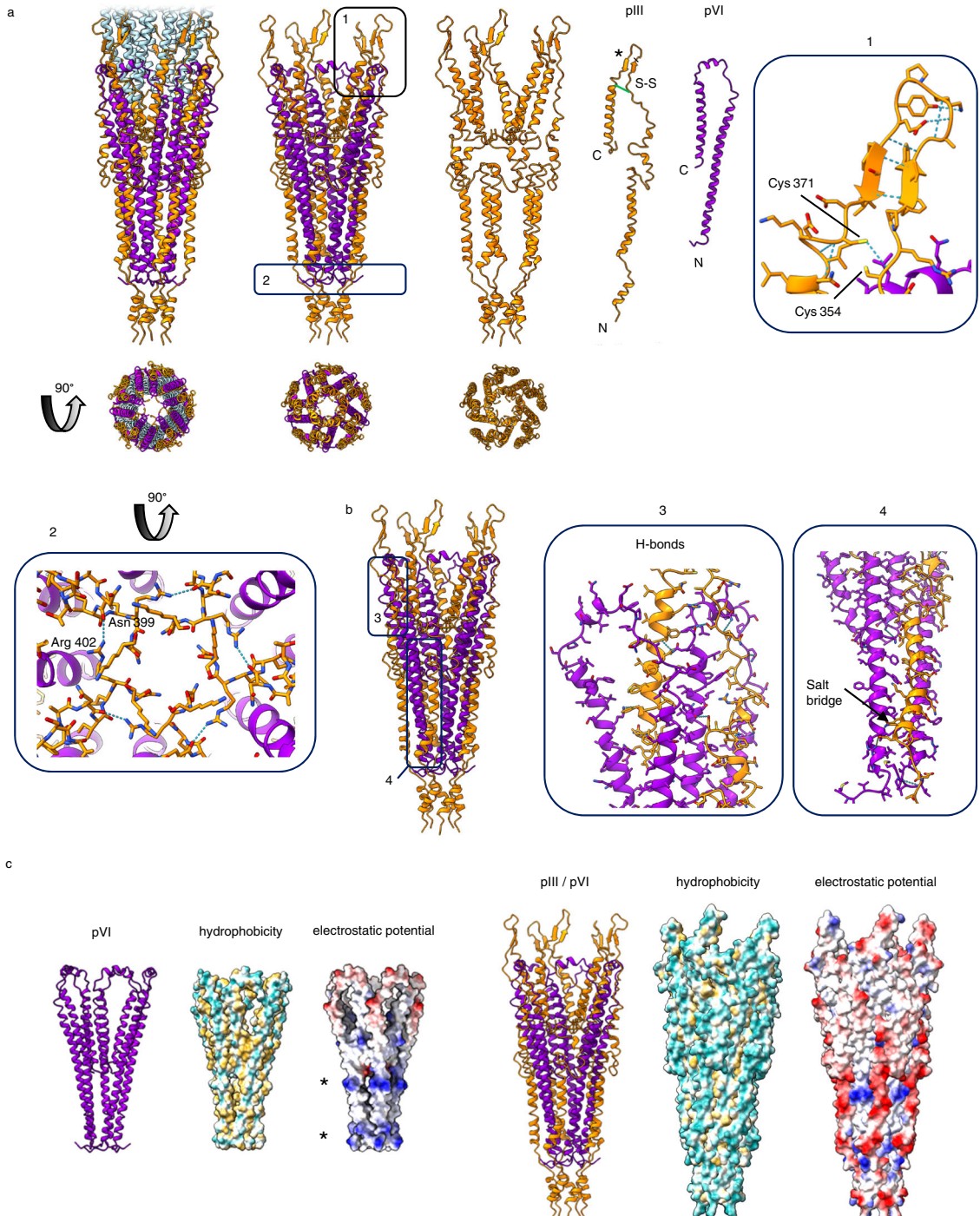

**Fig. 3 | Structure of the pointy tip. a** Structure of the pointy tip in ribbon representation with the C domain of pIII shown in orange, pVI in purple and pVIII in light blue. The tip is shown in two views 90° apart to show a side view and a view looking down the phage from the pointy tip. Proteins pIII and pVI are shown individually as well as in the phage assembled pentameric structure. Density for the pIII N1-N2 domains, plus the final 2 residues at the C-terminus, is not visible in our map and is not shown. The disulphide bond in pIII is shown in green and labelled S-S. The * highlights the β-hairpin loop. Regions 1 and 2 are boxed. Box 1 shows hydrogen bonding within the pIII β-hairpin (blue dashed lines) and the cysteine residues within the correct distance to form a disulphide bond. Box 2 shows a view looking down the phage towards the pointy tip, and a hydrogen bond formed between the sidechain of Arg 402 and the mainchain carbonyl oxygen of Asn 399 from a neighbouring pIII chain. pIII is shown as sticks, pVI as ribbons, hydrogen bonds as blue dashed lines. **b** Interactions between pIII and pVI. Regions 3 and 4 are boxed and show the hydrogen bonds (blue dashed lines) observed between one pIII molecule and two neighbouring pVI molecules, and the salt bridge between Glu 277 and Arg 12. **c** Pointy tip shown with pVI bundle only, and pIII/pVI bundle, in side view shown as ribbons, and as surface views of hydrophobicity and electrostatic potential. The most hydrophilic residues are shown in cyan and the most hydrophobic in mustard yellow; negative residues are shown in red and positive residues in blue. * denotes the rings of positive charge in pVI.

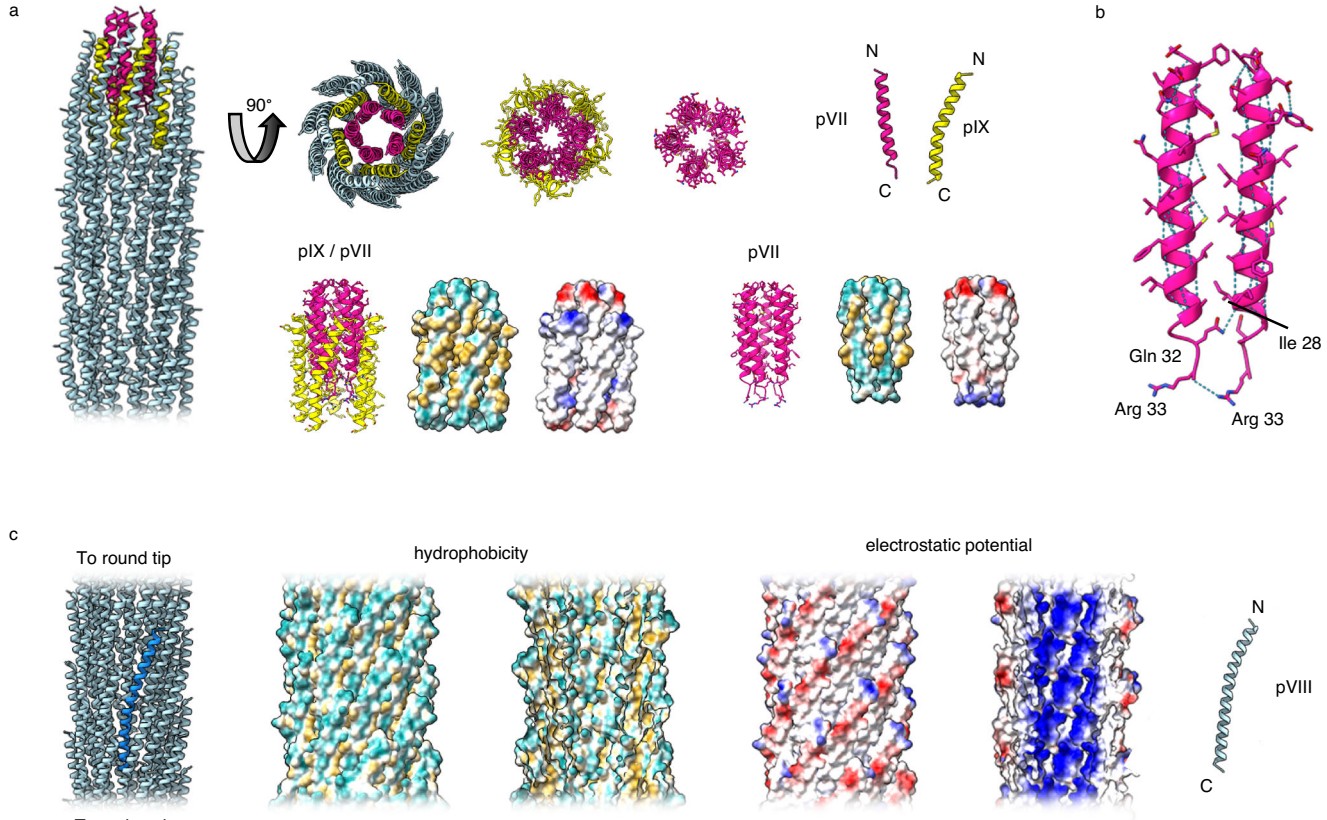

**Fig. 4 | Structure of the round tip and filament. a** Structure of the round tip in ribbon representation with pVII shown in pink, pIX in yellow and pVIII in light blue. The tip is shown in two views 90° apart to show a side view and a view looking down the centre of the phage from the round tip. Proteins pVII and pIX are shown individually, as pentamers, and in the complete structure. Side views are also shown as surfaces depicting hydrophobicity and electrostatic potential. The most hydrophilic residues are shown in cyan and the most hydrophobic in mustard yellow; negative residues are shown in red and positive residues in blue. Density for the 5 residue sequence at the far N-terminus of pVII (MEQVA, contributing to the negative charge and of mixed hydrophobicity) is not visible in our map and is not shown. **b** Hydrogen bonds in the pVII bundle, shown in ribbon representation. Two interchain hydrogen bonds are formed between neighbouring Gln 32 and Ile 28, and Arg 33 and Arg 33. **c** Structure of the pVIII filament shown, from left to right, as a ribbon diagram (one pVIII subunit highlighted in bright blue), with the outer surface coloured by hydrophobicity, inner surface coloured by hydrophobicity, outer surface coloured by electrostatic potential, inner surface coloured by electrostatic potential, and as a single protein chain as a ribbon. The most hydrophilic residues are shown in cyan and the most hydrophobic in mustard yellow; negative residues are shown in red and positive residues in blue. Density for the 4 residue sequence at the far N-terminus of pVIII (AEGD, overall negatively charged and hydrophilic) is not visible in our map and is not shown.

charged and hydrophilic. The pVIII helices are organised with their N-termini pointing towards the round tip, and their C-termini towards the pointy one (Supplementary Fig. 10d).

Overlaying a protomer of our f1-derived nanorod pVIII with a previously determined fd phage pVIII structure (2C0W; determined by X-ray fibre diffraction[26]) shows that the two are broadly similar (RMSD of 1.0 Å across 39 pruned pairs; RMSD of 1.5 Å across all 46 pairs; Supplementary Fig. 11). However, when comparing the pentamers by aligning at one protomer, small differences can be seen in the placement of the remaining four chains. The Ff family shares 40% sequence identity with filamentous phage Ike (Supplementary Fig. 11), with large differences in amino acid properties in some areas. However, overlaying a monomer of pVIII from Ike (6A7F)[37] with the f1-derived nanorod pVIII shows an even smaller difference between the helices with an RMSD of 0.7 Å (across all 46 pairs). When comparing the Ike and f1-derived nanorod pVIII pentamers, there are again small differences seen in the packing of the individual helices. Interestingly, Ike has a proline at position 30 while all Ffs have an alanine. This proline mid-way through the helix was proposed to cause the kink observed in the Ike structure[37], however, the kink is still present in the f1-derived nanorod in the absence of the proline.

## Structure of the assembled phage

Ffs can be thought of as pentameric building blocks with the first layer closest to the round tip (Fig. 5a). In layer 1, 10 helices of alternating pIX and pVIII line the outside of the pVII bundle (Fig. 5a). Interactions are mainly hydrophobic, with salt bridges between pIX and pVII subunits, and hydrogen bonds between pIX and pVIII subunits, adding further stability to the tip structure (Fig. 5a). Further copies of the main capsid protein pVIII continue to coat the round tip in two further layers (layers 2–3), interacting with the pIX helices by mostly hydrophobic interactions.

The presence of the major capsid protein pVIII in the round tip means that the few first layers of pVIII make unique interactions as the round tip section transitions into the filament (Fig. 5a). In layer 1, pVIII interacts with pVII, pIX and neighbouring pVIII molecules. In layers 2 and 3, pVIII interacts with pIX and neighbouring pVIII molecules. The pVIII in layer 4 onwards interacts with other pVIII molecules only. The helical symmetry of the filament is slightly altered as it comes close to the round tip, presumably because of the slightly altered binding partners, and the change from helical symmetry of the filament to only 5-fold symmetry at the very tip. In the filamentous section, pVIII interacts with its neighbours by mainly hydrophobic interactions, with

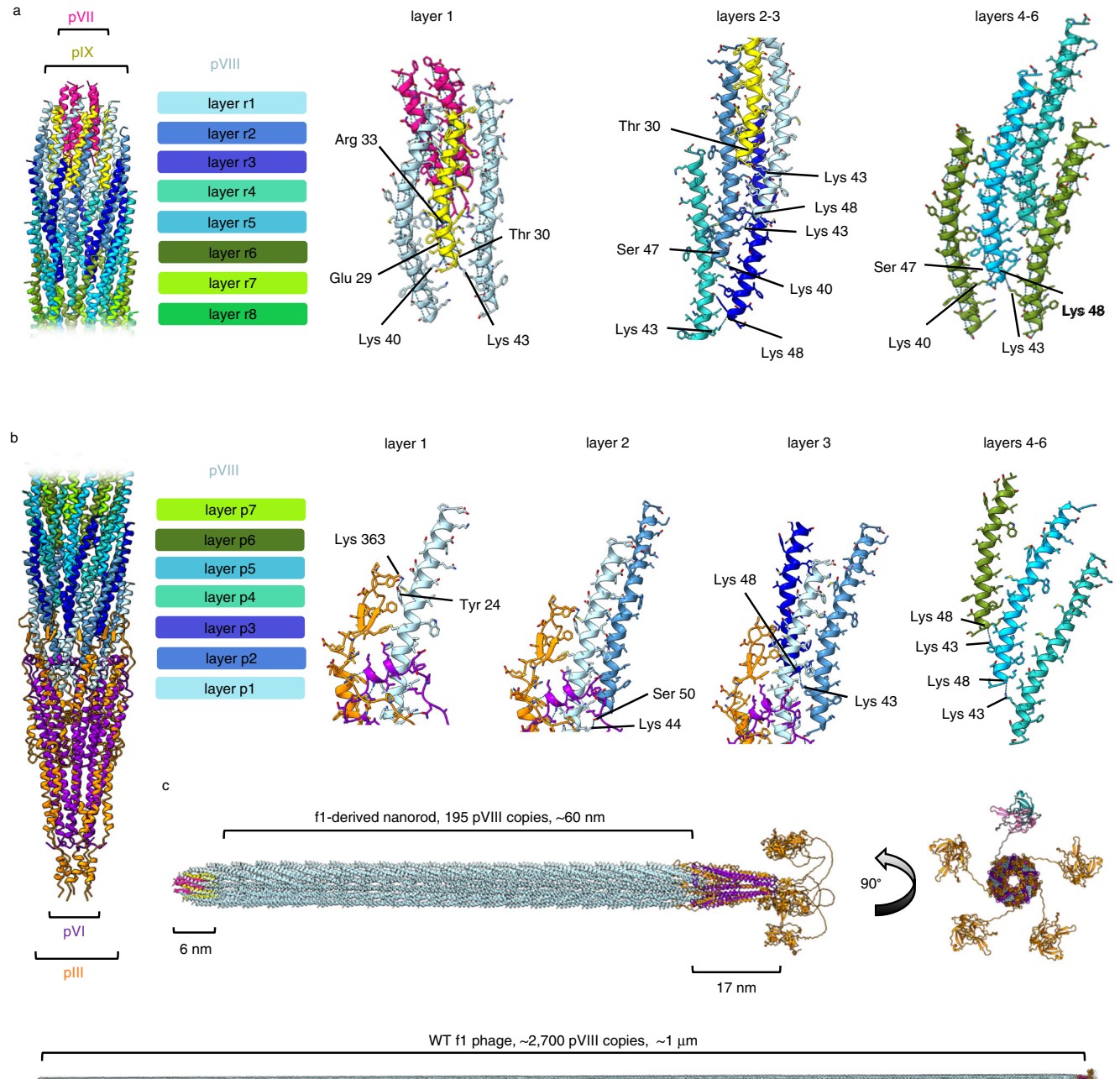

**Fig. 5 | Structure of the assembled phage.** Pentameric layers of pVIII are coloured differently according to their position in the filament e.g. layer r1 in light blue is the first layer at the round (r) tip, and layer p7 in bright green is the 7th layer at the pointy (p) tip. pVII is shown in pink and pIX in yellow. Hydrogen bonds within the different layers are shown with blue dashed lines. **a** Five copies of alternating pIX and pVIII line the outside of the pVII bundle at the round tip. In layer 1, interactions are mainly hydrophobic, with salt bridges between pIX and pVII (Glu 29 to Arg 33), and hydrogen bonds between pIX and pVIII (Thr 30 to Lys 43, and Glu 29 to Lys 40). In layers 2–3, pVIII interacts with pIX and neighbouring pVIII molecules via hydrophobic interactions. The pVIII in layer 4 onwards interacts with other pVIII molecules only, via hydrophobic interactions and with inter-subunit hydrogen bonds between Lys 43 and Lys 48, and Ser 47 and Lys 40. **b** Layers of pVIII interact with 5 copies of pIII and pVI at the pointy tip. In layer 1, pVIII interacts with pIII, pVI and neighbouring pVIII molecules, forming hydrogen bonds between Tyr 24 and Lys 363 of pIII, and between Lys 44 and Ser 50 of a neighbouring pVIII from layer 2. In layer 2, pVIII forms hydrogen bonds from Lys 43 to Lys 48 of neighbouring pVIII molecules from layer 3. In layer 4 onwards, pVIII interacts with other pVIII molecules only, with hydrogen bonds between Lys 43 and Lys 48 from adjacent layers. **c** Composite models of the f1-derived nanorod and WT f1 phage were generated by aligning the tips with multiple copies of the pVIII filament protein and an AlphaFold model of the linkers and N1-N2 domains of pIII. pVII is shown in pink, pIX in yellow, pIII in orange, pVI in purple and pVIII in light blue. A view rotated 90° depicts four of the N1-N2 domains of pIII in orange and one coloured differently to highlight the N1 domain (teal) and N2 domain (light pink).

hydrogen bonds observed at the C-termini and mostly involving lysine residues (Fig. 5a).

At the pointy tip, pentameric layers of pVIII interact with the tip proteins pIII and pVI (Fig. 5b). In layer 1, pVIII interacts with pIII, pVI and neighbouring pVIII molecules, with hydrogen bonds linking to layer 2.

The hydrogen bonding with pIII occurs in the β-hairpin loop that is stabilised by a disulphide bond (Fig. 3a). In layer 2, pVIII forms additional hydrogen bonds to pVIII molecules from layer 3 (Fig. 5b). In layer 4 onwards, pVIII interacts with other pVIII molecules only, with hydrogen bonds between adjacent layers.

## Model of the f1 filamentous phage

To visualise the structure of an assembled filamentous phage, we aligned multiple copies of the central, filamentous region with both caps (Fig. 5c). The N1-N2 domains of pIII, which have been observed previously as small knob-like structures connected to the virion by a string of flexible linker[38], were not visible in our map, and we did not observe any blurred patches of density in our 2D class averages which might have arisen from these domains (Supplementary Fig. 3a). Therefore, we used AlphaFold[39] to predict the structure of the linkers and N1-N2 domains. The software based the output on the available N1-N2 X-ray structures (1G3P, 2G3P)[14,34] (Supplementary Fig. 12). The mostly ß-stranded N1 and N2 domains form a horseshoe arrangement, with the extensive linker regions being disordered and therefore able to move freely. Although shown symmetrically (Fig. 5c, Supplementary Fig. 13, Supplementary movie 4), these domains could be found in a variety of different orientations surrounding the main body of the pointy tip.

The round tip measures 6 nm from the tip of pVII to the furthest part of pIX, thus comprising only 0.6% of the total filament (based on a 1 μm total length of WT phage, Fig. 5c). The pointy tip measures 17 nm from the top of the pIII β-hairpin loop to the furthest part of the C1 domain (without N1 and N2), thus comprising 1.6% of the total WT filament (Fig. 5c). The assembled phage displays a clear charge separation. In particular, the N domains and pointy tip of pIII are mostly negatively charged, and the phage lumen is overwhelmingly positive due to the C-terminus of pVIII (Supplementary Fig. 10d, Supplementary Fig. 13b). Both the phage surface and lumen are lined with hydrophilic residues, with a line of hydrophobic residues mediating interactions between the individual protein subunits (Supplementary Fig. 13c).

## Phage DNA

Density was visible for the single-stranded circular DNA genome. However, the density was not well defined and did not allow a detailed molecular model to be built (Supplementary Fig. 14a, b). At the round end, the 5 C-terminal arginine residues of pVII (Arg 33) and 5 arginine residues from pIX (Arg 26) form a positively charged ring that butts up to the tip of the DNA (Supplementary Fig. 14c), where the packaging hairpin is expected to be found[21]. The positively charged lumen of the virion is comprised mainly of lysine residues from the C-terminal ends of pVIII molecules (Supplementary Fig. 10d), hence allowing them to interact with the negatively charged DNA molecule. Four lysine residues from each pVIII monomer (Lys 40, 43, 44 and 48) line the DNA cavity (Supplementary Fig. 14d). We fitted a fragment of B-DNA into the density in our map to confirm the approximate dimensions of the nanorod circular ssDNA molecule (Supplementary Fig. 14e).

## The role of pIII in infection

pIII is responsible for receptor binding and subsequent infection of bacteria; our structure allows us to investigate the mechanism. Based on prior functional data, the pIII C domain was divided into C1, C2 and M (transmembrane) sub-domains by mutational analyses[23,40]. With knowledge of the structure, we now propose that the pIII sub-domain nomenclature follows the position of the α-helices and linkers (Supplementary Fig. 15a).

To examine the roles of specific structural features of the C domain in infection, a series of N1-N2 fusions to truncations of the C domain[41] were tested for their ability to complement phage containing a complete deletion of gene III (phage f1d3, Supplementary Fig. 15b, c). This analysis showed that phages with short truncations in the C domain (up to 24 residues) were as infective as the wild-type pIII-complemented phage (Supplementary Fig. 16, Supplementary Table 2). Mutants with increasingly larger C domain deletions (up to 52 residues) correlated with decreased infectivity by over 30-fold. The deletion mutant with the shortest C-terminal fragment (NdC83; 62 residue deletion) had infectivity equivalent to the negative control

phage that contained no pIII (infectivity is six orders of magnitude less than the wild-type pIII-complemented f1d3 phage). We mapped the truncations to the structure of the phage tip, which indicated that the C2 helix (a part of which was previously defined as the Infection Competence Sequence; ICS)[41] was essential for infection (Supplementary Fig. 16a).

Stability of phages can be assessed by investigating their resistance to detergents with different properties. Phages containing wild-type pIII were completely resistant to ionic detergents SDS and sarkosyl (Supplementary Fig. 17). In contrast, all phages with truncated C domains were completely disassembled in the presence of SDS, and phages with the largest truncations (i.e. that lacked most of C2) were additionally disassembled in the presence of sarkosyl, a less polar detergent with a larger head group. Interestingly, the predicted octanoic and butanoic acid molecules (Supplementary Fig. 8) would likely be exposed in all the truncated pIII mutants. Accessibility of lipids to detergent could explain the change in phage stability; in this case explaining why sarkosyl (with the larger head group compared to SDS) only affects the stability of the mutants with the largest truncations. This analysis highlights the importance of the C2 domain in infection and stability of phage, and is corroborated by the structure, where the complete C2 helix would shield the hydrophobic pVI core from external solutes (Fig. 3c).

## Membrane integration

It has been demonstrated that the major capsid protein pVIII integrates into the cytoplasmic membrane on infection[18] and is recycled in the assembly of new phage particles[42]. It has also been suggested that the minor capsid tip proteins pVII and pIX are reused for new virions[11], whereas pIII and pVI are not[22]. Being as all 5 capsid proteins exist as cytoplasmic membrane proteins prior to assembly into new phage particles[20], it is expected that all of them integrate into the cytoplasmic membrane on infection. We therefore investigated the predicted transmembrane regions of all 5 proteins to understand more about the conformational changes that would need to occur on transitioning from and to the phage assembled state (Supplementary Fig. 18). Transmembrane sequence analysis with MEMSAT-SVM[43] predicts that pIII has one transmembrane helix and pVI has 3 (Supplementary Fig. 18a). pVII, pVIII and pIX are all predicted to have one transmembrane helix each (Supplementary Fig. 18a). All capsid proteins are predicted to be oriented with their N-termini in the periplasm and their C-termini in the cytoplasm. Proteins pIII and pVI would need to undergo significant conformational changes to transition from their structure in the phage particle to the membrane-embedded state (Supplementary Fig. 18b, d). The N-terminal domain of pIII would need to swing out around the β-hairpin loop, and the long α-helix of pVI would need to form two shorter helices. Interestingly, AlphaFold predicted that pVI could fold up into a more compact 4-helix form (Supplementary Fig. 18c).

## Discussion

Based on the structure of the f1-derived nanorod, it is possible to reconcile a wealth of phenotypic data and refine current working models for the mechanisms of phage binding, infection, egress and release (Figs. 6 and 7).

The pIII tip is responsible for receptor binding on bacteria−first to the tip of the extracellular F-pilus (via the N2 domain)[13], and second to the TolA protein anchored in the cytoplasmic membrane (via the N1 domain)[44]. The N1 and N2 domains of pIII are bound tightly to each other in a horseshoe shape[14,34] (Supplementary Fig. 19a). The N2 binding site on the N1 domain overlaps with the TolA binding site[15] (Supplementary Fig. 19b, c). Therefore, F-pilus binding must unfold the N1-N2 hinge to expose the TolA binding site on N1[45,46] (Supplementary Fig. 19c, d, Supplementary movie 5). Less than 5 copies of N2 bound to the F-pilus are sufficient for infection[17,41], yet the long glycine-rich

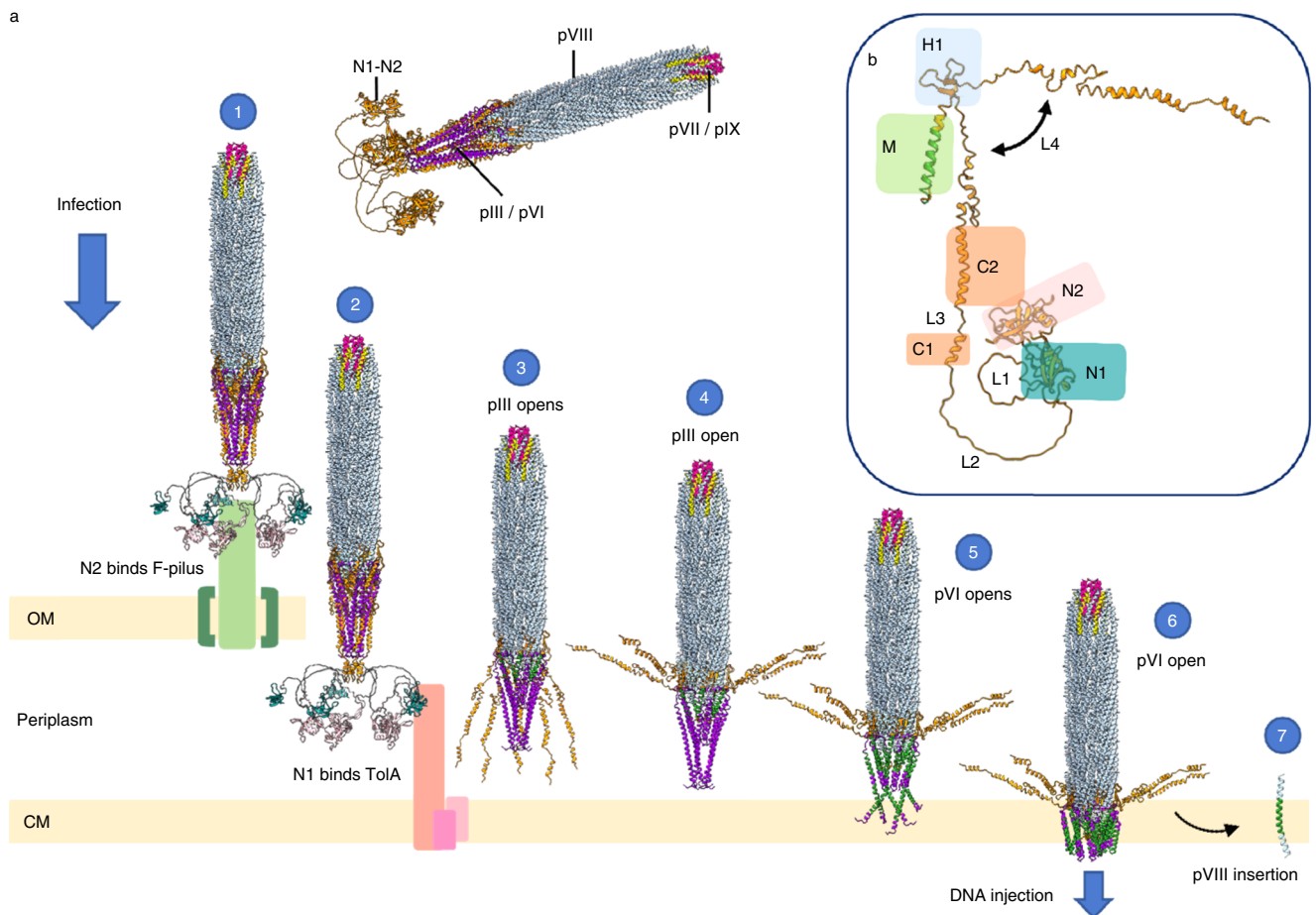

**Fig. 6 | Working model for infection of the Ff family of phages.** The phage used in the model contains 65 copies of the filament protein pVIII; WT phage would be ~26x longer. pVII is shown in pink, pIX in yellow, pIII in orange, pVI in purple and pVIII in light blue. Selected pIII N1 domains are coloured teal and N2 domains light pink. OM, outer membrane; CM, cytoplasmic membrane. **a** Model for infection with **b** domains of pIII highlighted. (1) The N2 domain of pIII binds to the F-pilus and unfolds the N1-N2 hinge to expose the TolA binding site on N1. It is unknown how many N2 domains are needed to bind to the F-pilus for high-efficiency infection. (2) The released N1 domain of pIII binds to the TolQRA complex. (3) The C1 and C2 domains of pIII start to move away from the body of the phage at the H1 β-hairpin loop due to flexibility in the L4 linker (**b**). The N1-N2 domains are not shown for clarity but could remain bound to TolQRA. The pIII transmembrane (M) helix is coloured green. (4) Complete opening of the C domain of pIII exposes the hydrophobic pVI pentamer. (5) The exposed hydrophobic regions of pVI insert into the membrane (transmembrane domains coloured green), also allowing insertion of the now exposed pIII M domain (**b**). (6) The pore formed by a pentamer of pIII and a pentamer of pVI allows DNA injection into the cytoplasm. (7) The first layer of pVIII is exposed on loss of the pointy tip, and proteins move laterally into the bilayer due to opposing charge on the N and C-termini. This reveals the second layer of pVIII, and the process continues.

linkers between the pIII C domain and N1-N2 could allow up to 5 molecules of pIII to clamp around the filament for high-efficiency infection (Supplementary Fig. 19d, Fig. 6a).

The high-resolution structure of a type IV secretion system[47] (which assembles a conjugative pilus related to the F-pilus[48]) reveals an intriguing similarity between the stalk structure (VirB5/TrwJ), suggested to locate at the tip of the assembled filament[47], and the pIII C domain at the tip of f1 (Supplementary Fig. 20a). The identity of the protein (or proteins) at the tip of the F-pilus is not known currently. However, it is interesting to observe that both VirB5/TrwJ and pIII are comprised of pentamers of α-helices, arranged symmetrically, with the same diameter across and opposing charge (Supplementary Fig. 20b). It is possible that a similar shape and charge complementarity at the tip of the F-pilus would allow it to interact with the tip of f1.

F-pilus retraction brings the phage to the bacterial outer membrane[5]. The type IV secretion system structure suggests that the dimensions of the outer membrane complex are sufficient to accommodate a pilus[47]. f1 is narrower than the F-pilus[49] (Supplementary Fig. 20c); it is therefore possible that phage can cross the bacterial outer membrane by passing through the F-pilus assembly machinery. Intriguingly, however, phage can infect cells without F-pili in the presence of

$Ca^{2+}$ ions, albeit at 6 orders of magnitude reduced efficiency[50]. It therefore cannot be excluded that the role of binding to the F-pilus is to concentrate phage at the outer membrane where crossing of the bilayer occurs by an unknown mechanism. Once the phage tip has crossed the outer membrane, the N1 domain of pIII can locate and bind to the periplasmic C or III domain of TolA that is located close to the outer membrane, pulling the phage into the periplasm (Fig. 6a).

Phage must next navigate the cytoplasmic membrane. No bacterial proteins other than the TolQRA complex have been implicated in the process, thus it is likely also mediated by the leading pIII tip. It has been suggested that a conformational change occurring on N1 binding to TolA can initiate a loosening of the interactions between pIII helices in the C domain of the pointy tip[17]. TolQ and TolR could also be involved indirectly (by forming the TolQRA complex) or directly in these conformational changes[51]. In support of the involvement of the C domain of pIII in infection, we have demonstrated that the C2 domain is essential for both infection and stability of the phage particle. The L2 linker between the pIII C domain and the N1-N2 domains (Fig. 6b, Supplementary Fig. 15a) would be sufficiently long (up to 14.8 nm) to allow the C2 domain to reach the transmembrane part of the TolQRA complex in the cytoplasmic membrane, even if the N1-N2 domains

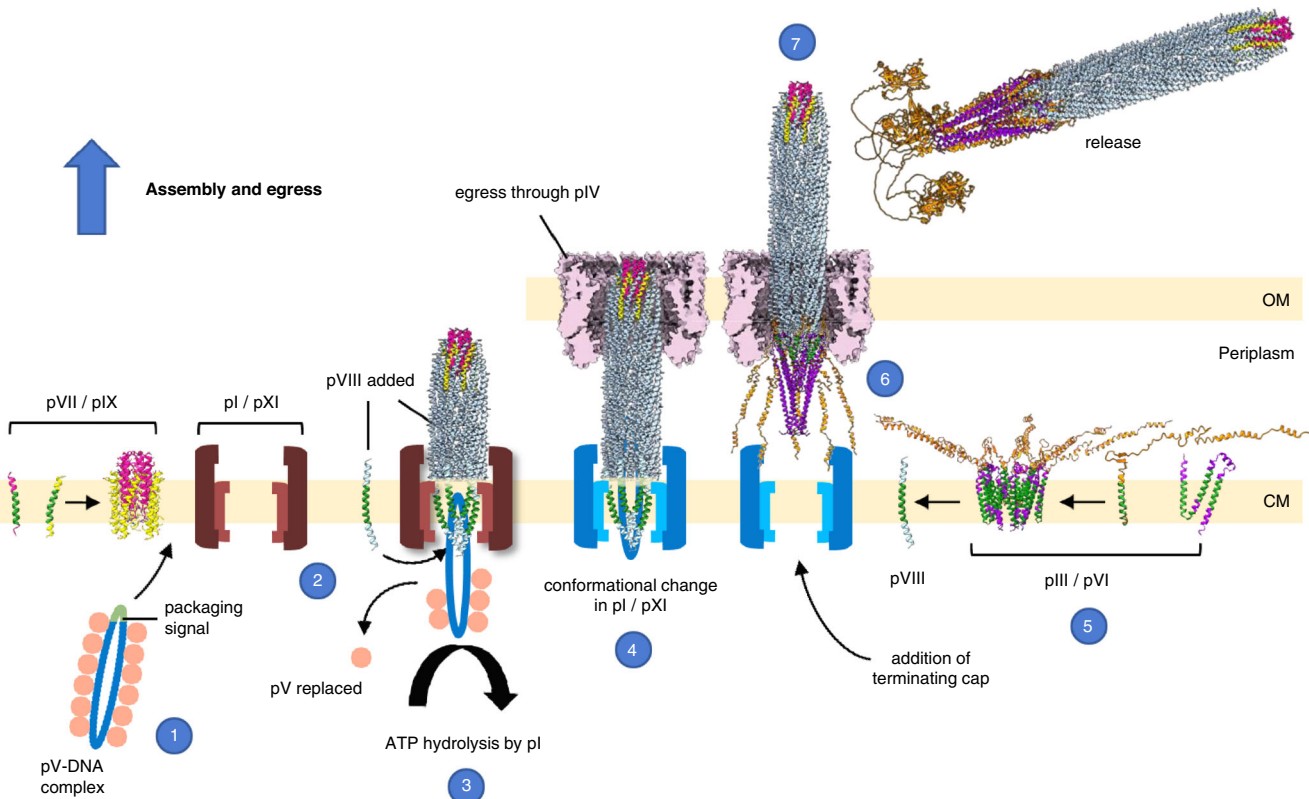

**Fig. 7 | Working model for assembly and egress of the Ff family of phages.** The phage used in the model is depicted as per Fig. 6. OM, outer membrane; CM, cytoplasmic membrane. (1) ssDNA is packaged by pV, leaving the packaging signal hairpin exposed (green). (2) pV targets the DNA-protein complex to pI/pXI, where pV is replaced by the pVII/pIX round tip. Displacement of pV allows pVIII to diffuse laterally around the DNA. (3) ATP hydrolysis by pI/pXI drives growth of the filament with continuous addition of pVIII. (4) On reaching the terminus of DNA, pI/pXI

undergoes a conformational change which increases affinity for the pIII/pVI tip, which forms a membrane-bound complex (5). (6) The L4 linker closes around the body of the phage at the H1 β-hairpin loop (Fig. 6b) and the C domain of pIII folds down over the hydrophobic helices of pVI to secede the phage from the cytoplasmic membrane. (7) Phage passes through the pIV secretin complex (mauve) in the outer membrane and is released. The flexible linkers and N0 domains aligning pIV with the pI/pXI complex shown in Fig. 1 have been omitted for clarity.

remain bound to their receptors, as has been predicted previously[44]. This idea is in agreement with the fact that the distance between the N1-N2 domains and the C domain must be maintained for high levels of infectivity[52]. It could therefore be envisaged that the pentameric helices of the pIII C domain (most of C1 and C2) are prised apart by binding TolA in the periplasm, so exposing the hydrophobic pVI pentamer (Supplementary movie 6). The flexible L4 linker between the M and C2 domains of pIII could allow for a twisting out of C1 and C2 about the β-hairpin loop (H1) relative to the transmembrane (M) helix (Fig. 6b). Without pIII surrounding pVI and stabilising it, the hydrophobic regions of pVI would become exposed, causing pVI to revert to its 4-helix form and insert into the membrane (Fig. 6a, Supplementary movie 6). The loss of extended pVI would alter the binding partners of the remainder of pIII (up to the β-hairpin structure), now revealing the C-terminal M domain of pIII. This idea is consistent with the fact that pIII has been shown to form a pore in liposomes[53], and AlphaFold predicts the pVI pentamer, and also a pentameric pIII/pVI complex, as channels with sufficient dimensions to accommodate DNA (Supplementary Fig. 21a–c). It is interesting to note that the open state tip complex is reminiscent of the family of pentameric ligand-gated ion channels[54]. These are also pentamers of transmembrane α-helices, with an extracellular domain containing a distinctive cysteine-loop motif, like pIII. When a ligand binds to the extracellular domain of the ion channels, a conformational change occurs, causing the helices to move apart[54]. It is therefore plausible that a similar mechanism of channel opening allows a pore of pVI and the M domain of pIII to open, allowing injection of DNA into the cytoplasm (Fig. 6a).

The major capsid protein pVIII is inserted into the cytoplasmic membrane on infection[18]. Transmembrane helices tend to orient with positive residues in the cytoplasm, and negative on the opposing side[55]. The two termini of pVIII have opposing charges, with the N-terminus being mostly negative, and the C-terminus highly positive (Supplementary Fig. 10d). After opening of the pointy tip, the pVIII charge distribution would thus aid insertion of each helix into the bilayer, via detachment from the main body of the phage and lateral diffusion into the membrane (Fig. 6a). Finally, the terminal pVII and pIX in the round tip would reach the cytoplasmic membrane, where they would also diffuse laterally into the bilayer due to the presence of a hydrophobic segment each (Fig. 6a).

All phage capsid proteins are embedded in the cytoplasmic membrane prior to assembly[20]. It is therefore likely that the conformational changes needed for assembly of proteins into the phage will be similar to the reverse of those occurring during infection. Phage assembly requires ATP hydrolysis provided by the pI component of the pI/pXI complex, and a proton motive force[56]. The DNA is first packaged by the phage-encoded protein pV, which wraps around the DNA in a rod of dimers, leaving the packaging signal hairpin exposed[57] (Fig. 7). pV targets the DNA-protein complex to the pI/pXI ring-shaped multimer in the cytoplasmic membrane[58] where the DNA hairpin structure is thought to act as a signal for replacement of pV by the pVII/pIX round tip[21] (Fig. 7). The positively charged pocket observed inside the pVII/pIX tip in our structure (Supplementary Fig. 14c) could feasibly mediate the interaction with the DNA hairpin. This is corroborated by the fact that the C-terminal residues of pVII and pIX have been found to

interact with phage DNA[21]. Binding could cause the displacement of pV and allow pVIII to diffuse laterally around the DNA at the cytoplasmic membrane, via the positively charged C-terminus interacting with the negatively charged phosphate groups of DNA[59]. ATP hydrolysis by pI/pXI would subsequently drive growth of the filament (Fig. 7).

Termination occurs by addition of the pIII/pVI pointy tip; both of these proteins are required for the release of filamentous phage from the infected cells[22]. It has been suggested that the pI/pXI complex undergoes a conformational change on reaching the terminus of the DNA, which increases the affinity for the tip proteins[60] (Fig. 7). Given that our structure shows that the helices of pIII and pVI are tightly intertwined, it is plausible that pIII and pVI form a membrane-bound complex prior to assembly into the phage particle. This is supported by the findings that pIII protects pVI from proteolytic degradation in *E. coli*[22]. It has also been shown that pIII and pVI each associate with the filament protein pVIII prior to assembly into the phage[20] (Fig. 7).

The structures of pIII and pVI support a model in which pIII needs to fold over the hydrophobic C-terminal transmembrane helices of pVI in order to secede from the cytoplasmic membrane. The residues of pIII necessary for co-integration of mutants into the virion together with the wild-type pIII[61] have been mapped onto our structure and can be seen to lie within the transmembrane (M) helix, the β-hairpin loop (H1) and the L4 linker (Supplementary Fig. 22). Mutation of the cysteines in the β-hairpin loop of pIII results in phage that cannot assemble[62], highlighting the importance of stability in the predicted hinge region (Fig. 6b, Supplementary Fig. 22). The residues essential for assembly therefore all lie within a 93-residue stretch corresponding to the C-terminus of pIII including the L4 linker. The C1 domain, L3 and C2 domain are not essential for assembly. It is plausible that the H1-L4 region is key for the conformational changes that would need to occur, causing the L4 linker region to swing down to the closed state around the hinge. With respect to termination, a pIII C domain truncation that is 83 residues long cannot terminate assembly, but a fragment 10 residues longer can[23]. Mapping these 10 residues onto our structure locates the key amino acids required for termination in the small loop where C2 joins the L4 linker (Fig. 6b, Supplementary Fig. 22). For the virion to be released from the cells, the M domain of pIII will need to be extracted from the membrane, together with pVI, for assembly into the phage particle. It is plausible that pIII needs to be of a sufficient length to secede the pVI hydrophobic helices from the membrane, disrupting the hydrophobic interactions with the phospholipid bilayer, pVIII, and/or the pI/pXI transmembrane complex, resulting in release.

Mixing different pIII mutants within a virion has been reported to affect entry and release differently. For example, the complete C domain (only lacking N1-N2) was shown to complement the assembly deficiency of the NdC83 internal deletion mutant, but not its lack of infectivity[17]. Furthermore, it was shown that NdC83 has a dominant-negative effect on infectivity of the virion when combined with the wild-type pIII[63]. These two observations differentiate triggers and conformational transitions involved in Ff entry and release, which become examinable now that the structure of the pointy tip has been solved. Practically, combinations described above have been the basis for a major improvement in the power of phage-assisted continuous evolution (PACE)[63].

In phage display, protein domains or peptides are often fused to the N-termini of either the full-length pIII or the pIII C-terminal domain[64]. However, it has also been shown that peptides can be linked to the C-terminus of pIII, which we visualise as buried in our structure. Space limitations in the pIII central stricture would explain why only an extra 9 residues are tolerated in this position[23]. Longer peptides fused to the pIII C-terminus can only be displayed on the surface of the phage if combined with the wild-type pIII in the same virion, and if a glycine linker is used[65]. In this case, the linker must pass between helices in the tip, to allow the peptide to become exposed on the exterior of the phage particle.

Near-atomic resolution structures of filamentous phage tips in conjunction with structure-function analyses is paradigm-shifting for virology. We reveal how an intertwined network of α-helices form a highly stable filament, culminating with a pentameric bundle at the leading tip of the infecting phage. This knowledge allows the mechanism of cellular-attack to be rationalised and can now be exploited for mechanistic understanding of the infection and assembly/egress of all filamentous phages. Minor Ff capsid proteins demonstrate great plasticity with respect to incorporation of truncated and mutated subunits into virions, as well as tolerance to protein fusions of a broad size range[64,66]. Structural details of the tips will undoubtedly enable improvement of phage display by combining modified virion proteins. In addition, the structures will allow expansion of biotechnological and nanotechnological applications by allowing the precise structure-guided design of novel modification points and 3D display structures.

## Methods

### Generation and purification of the f1-derived nanorods

The nanorods containing a 529-nt circular ssDNA backbone were produced using a two-plasmid system. A helper plasmid (pHP1YM; *IR1-B(C143T); gVIII4ᵃᵐ*, pVIII Y21M) encodes all f1 proteins, and the nanorod template plasmid (pBSFpn529) contains a replication cassette that generates the 529-nt circular ssDNA (details in Supplementary Fig. 1). The nanorods were produced from 1 L of pooled double-transformed host cells. After overnight incubation of the transformed cell culture, nanorods were purified from the supernatant and concentrated by PEG using standard protocols[67], with the exception of an increased concentration of PEG due to the short length of the nanorods (15% PEG, 0.5 M NaCl). Furthermore, the nanorod solution after the first PEG precipitation was treated by DNAse and RNAse to remove the cell-derived DNA and RNA. Concentrated nanorods were purified by CsCl density gradient centrifugation using standard Ff (f1, fd or M13) protocols[67]. Further purification was achieved by anion exchange chromatography using a SepFast™ Supor Q column (details in Supplementary Fig. 2). pVIII contains a Y21M replacement shown previously to result in identical conformation of all copies of the major coat protein[26]. Plasmids are available from J. Rakonjac upon request.

Quantification of nanorods was performed by densitometry of the ssDNA. Briefly, nanorods were disassembled by incubation in ¼ volume of SDS-containing buffer (1% SDS, 1x TAE, 5% glycerol, 0.25% BPB) at 100 °C for 20 min. Disassembled nanorods, along with the quantification standard (serial dilutions of a 529-nt ssDNA of known concentrations purified from the nanorods), were analysed by electrophoresis on 1.2% agarose gels in 1x TAE buffer. All quantification samples were loaded in triplicate. The gel was stained in ethidium bromide, de-stained and imaged using a GelDoc™. Images were analysed using ImageJ[68]. A second-order polynomial function was used to fit the standard curve generated from the ssDNA standards and used to determine the number and concentration of nanorods.

### Cryo-electron microscopy

**Sample preparation.** 3 μl of 529-nt nanorods ($1.01 \times 10^{15}$ particles/ml) were applied to glow-discharged R1.2/1.3 Cu 300 mesh grids (Quantifoil) and frozen on a Mark IV Vitrobot (Thermo Fisher Scientific) with the following conditions: 4 °C, 100% relative humidity, wait time 5 s, drain time 0 s, blot force 0, blot time 4 s.

**Imaging.** Grids were screened using a 200 kV Talos Arctica microscope (Thermo Fisher Scientific) with a K2 Summit direct electron detector at the GW4 Regional Facility for CryoEM in Bristol, UK. A preliminary dataset was recorded and used to determine initial helical parameters. Micrographs used for the final structure were collected on a 300 kV Titan Krios microscope (Thermo Fisher Scientific) with a K3 Bio-Quantum direct electron detector (Gatan) at the Electron Bio-imaging Centre (eBIC) at Diamond Light Source, UK. Data were collected using

EPU software (Thermo Fisher Scientific) with a defocus range from −1.3 to −2.5 µm in 0.3 µm increments. The total dose was 40 electrons/Å$^2$ at a magnification of 81 kx, corresponding to a pixel size of 1.10 Å (0.55 Å super-resolution). Further details are shown in Supplementary Table 1.

**Data processing.** For the tips, motion correction, CTF estimation and particle picking were performed using Warp[30] with Box2Net trained using manual picking of ~1000 particles, using both round and pointy tips. 594,894 particles were picked and then imported into cryoSPARC 3.2.0[31]. Multiple rounds of 2D classification were used to separate the tips, followed by 3D classification. A total of 86,242 particles of the pointy tip and 255,372 particles of the round tip were selected for final refinement. Iterative 3D refinements and CTF refinements were run for both tips, resulting in resolutions of 2.9 Å and 2.8 Å for the pointy and round tips, respectively (FSC = 0.143). A map of the pointy tip was also generated without any symmetry imposed to check that the densities modelled as fatty acids were not symmetry-related artefacts. To attempt to visualise phage DNA, maps were calculated for both tips without symmetry applied. This was attempted with both a typical mask around the whole tip and also with a tight mask surrounding only the DNA. Approaches using particle subtraction and symmetry expansion with 3D classification were also attempted. The resulting reconstructions did not result in improved visualisation of the phage genome.

To process the central filament, cryoSPARC was used for motion correction and CTF estimation. The filament tracer tool was used to pick the particles, with an inter-particle spacing of ~29 Å. A total of 13,430,601 particles were picked, extracted and processed using 2D classification. A total of 1,201,228 particles were selected and used for further refinement. Helical parameters were determined using data from the initial Talos dataset. An initial helical refinement with no symmetry or helical parameters was run and used as a reference for a second refinement with C5 symmetry and no helical parameters. From this map, helical parameters were predicted using the symmetry search tool. The resulting rise and twist values predicted were 16.56 Å and 37.41 degrees, respectively. Helical refinements were performed on the Krios dataset using these values as a starting point, with CTF refinement and additional 2D classification performed. The helices in the final map overlaid well with the map without any symmetry applied. The final resolution achieved was 2.6 Å using 1,139,813 particles, with helical rise and twist values of 16.599 Å and 37.437 degrees, respectively, and C5 symmetry applied.

**Modelling and protein structure prediction.** The system of amino acid numbering used throughout is for the mature protein sequences of pIII and pVIII (i.e. the sequences are numbered from 1 after their signal sequences have been removed). A fibre diffraction structure of fd phage pVIII protein (PDB 2COW)[26] was first manually placed in the filamentous cryoEM map using UCSF ChimeraX[69]. Proteins pIII and pVI were easily distinguished from each other on the basis of their length, and were initially built into the pointy density map as poly-Ala chains using Coot[70] and then their sequences assigned manually. Proteins pVII and pIX were harder to distinguish on the basis of their length; they were also initially built as poly-Ala helices in the round map and their sequences assigned manually. For all three structures, bulky residues, glycine residues and unique sequence patterns were used to guide sequence assignment during model building. Model building and adjustments were performed using Coot, and the monomeric structures refined using Refmac[71] from the CCPEM suite[72]. 5-fold and helical symmetry were applied using ChimeraX, and the three complete structures refined again with Refmac. Models were validated using tools in Coot[70], Molprobity[73] and the RCSB validation server[74]. DeepEMhancer[75] was used for denoising and postprocessing of the maps, and these maps used to aid model building. Model measurements were taken, and figures prepared using ChimeraX[69]. A 12mer of B-DNA was created in Coot, and modelled in the density using the same software.

AlphaFold2[39] was used for protein structure predictions, with monomers and multimers predicted. MEMSAT-SVM[43] was used to predict transmembrane regions of the phage proteins via the PSIPRED server[76]. AlphaFold model 0 (Supplementary Fig. 12a) was used for the linkers and N1-N2 domains of pIII in the composite model of the f1-derived nanorod (shown in Figs. 5c, 6, 7, Supplementary Figs. 13, 15a, 19a, Supplementary movie 5). The N1-N2 domains were manually swung down relative to the C domain of pIII via the flexible linkers to model the position of the N1-N2 domains on F-pilus binding (Fig. 6a, Supplementary Figs. 19d, 20c, Supplementary movie 4).

**Functional analysis of the C domain.** Methods for production of C domain mutants of phage, testing infectivity and stability, are described in Supplementary Methods.

**Reporting summary**
Further information on research design is available in the Nature Portfolio Reporting Summary linked to this article.

## Data availability
The 3D cryoEM density maps generated in this study have been deposited in the Electron Microscopy Data Bank (EMDB)[77] under accession codes EMD-15831, EMD-15832 and EMD-15833 for pointy, round and central filamentous maps, respectively. The atomic coordinates have been deposited in the Protein Data Bank (PDB)[74] under accession numbers 8B3O, 8B3P and 8B3Q. The source image data used in this study have been deposited to the Electron Microscopy Public Image Archive (EMPIAR)[78] under accession number EMPIAR-11480. The previously determined structures of the fd pVIII (2COW), N1 domain of pIII bound to TolA (1TOL), N1-N2 domains of pIII (1G3P), the F-pilus (5LER), the stalk of the type IV secretion system (7O3V) used in this study are available in the PDB under accession codes listed. Uncropped versions of the nanorod purification gels (Supplementary Fig. 2), native agarose gel electrophoresis (Supplementary Fig. 17) and the data corresponding to the infectivity of pIII C domain mutants (Supplementary Table 2) are shown in the Source data file provided with this paper.

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

## Acknowledgements

This work was funded by a Wellcome Seed Award in Science awarded to V.A.M.G. (210363/Z/18/Z), which along with the University of Exeter, supported R.C. M.M. was supported by a BBSRC responsive mode grant awarded to V.A.M.G. (BB/R008639/1). Work at Massey University and Nanophage Technologies was supported by a Callaghan Innovation Grant (BNANO2101), Palmerston North Medical Foundation, School of Natural Sciences and donations by Anne and Bryce Carmine and an Anonymous Donor. B.D. received funding from the European Research Council (ERC) under the European Union's Horizon 2020 research and innovation pro-gramme (acronym Microrobots, grant agreement No. 803894). We acknowledge Diamond Light Source for access and support of the cryoEM facilities at the UK's national Electron Bio-imaging Centre (eBIC) at Diamond Light Source [under proposal BI25452], funded by Wellcome, MRC and BBRSC. We acknowledge access and support of the GW4 Facility for High-Resolution Electron Cryo-Microscopy, funded by Well-come (202904/Z/16/Z and 206181/Z/17/Z) and BBSRC (BB/R000484/1). We are grateful to Ufuk Borucu at the GW4 Regional Facility for assistance with screening, and to Trevor Loo, Massey University, for technical sup-port with FPLC. We are indebted to George P Smith (University of Missouri) and Marjorie Russel (Rockefeller University) for critical reading and comments on the manuscript.

## Author contributions

R.C. prepared samples for cryoEM, built the atomic models and analysed the structure. M.M. collected and processed cryoEM data to determine the structures. R.I.L.Q. constructed the nanorod production system, generated and purified the f1-derived nanorods. N.B. conducted func-tional analysis of the C domain mutants. B.D. interpreted data and pro-vided resources for electron microscopy. J.R. and V.A.M.G. conceptualised the project, designed the research and obtained the funding. V.A.M.G. wrote the manuscript with R.C. and J.R.; all authors commented on the manuscript.

## Competing interests

The authors declare no competing interests.
