## [Peer Review File · Nature Communications]

REVIEWER COMMENTS

Reviewer #1 (Remarks to the Author):

Conners et al report the first near-atomic resolution structure of a miniaturised f1 phage virion (nanorod) determined by cryoEM. f1 belongs to the filamentous phage family, which has been extensively studied from the fundamental standpoint. This groundwork allowed for a number of applications in research and biotechnology. Filamentous phages, which replicate without lysing their bacterial host and disseminate their DNA in the form of episomes, are also important drivers of horizontal gene transfer.

Mastering both the cryoEM analysis and the genetics of f1 enabled the authors of this work to exploit extensively the structural data and to derive plausible models of phage entry and egress. The study will be instrumental to reach a more complete understanding of these complex processes. The work offers an exciting possibility to tie the structural data to the wealth of genetic and biochemical data accumulated over the years on these phages and their assembly systems. The paper is very well written and illustrated, in particular using movies that provide plausible models for conformational changes that take place to allow for DNA packaging and injection. The cryoEM data are further supported by new experimental assessment of the roles of pIII protein regions and domains in phage infectivity and stability.

In summary, this is a very nice and complete study that represent a very important advance in phage biology. I have a few minor questions and comments below.

In the model describing phage assembly and egress depicted in Fig. 1, it is unclear what allows for the inner membrane assembly complex to align with the pIV secretin for efficient egress. The recent Conners et al publication on the pIV (ref 19) proposes the role for the N0 domain of pIV and the Fig. S1 depicts a purple circle and a black line connecting the IM complex to pIV. However, there is no mention of this feature in the figure legend nor in the scheme itself.

Line 117. The N1 and N2 domains of pIII are not visible and the predictions using alpha fold are poor. What is the predicted disorder score of the linker regions? Would they be vulnerable to proteolysis in the periplasm, and is it possible that due to their negative charge they bind calcium and are compacted in this compartment prior to egress?

Line 173. ... termini oriented...

Reviewer #2 (Remarks to the Author):

This is a beautiful paper that significantly advances studies of filamentous phages, something that has been the focus of structural studies for more than 50 years. The results are likely to have an impact on both biotechnology and work in the area of phage therapy. The approach taken to reconstruct an entire filamentous phage has been innovative, and most importantly, worked. I had only the most minor of points that the authors should be easily able to address in a slightly revised paper.

Line 105) “All structures had 5-fold symmetry, with the central filamentous region having additional helical symmetry.” This was quite confusing to me. The outer capsid certainly has helical symmetry in addition to the C5 point-group symmetry. Do they mean here that they only imposed C5 symmetry for an outer capsid map (and did not impose helical symmetry), and did not impose the C5 symmetry on the inner DNA density (but did impose helical symmetry)?

Line 108) “resolutions of 2.97 Å, 2.58 Å and 2.81 Å”. Do the authors really believe in the significance of citing resolutions to one-hundredth of an Å given that: a) one can easily change these values by making changes in the mask used; b) the FSC function is rarely monotonic; c) the “gold standard” FSC is not actually measuring resolution, but rather reproducibility.

Lines 120-121) It would be useful if more information were provided about the disulfide bond, rather than just saying that the two were in close enough proximity to form a disulfide. Was density seen for the disulfide? If not, could this be due to the extreme sensitivity of disulfides to electron irradiation (e.g., Pieri et al., PNAS 119, 2022)?

Line 156) “lipids are not uncommon in the capsids...”. It might also be mentioned that lipids have been observed tightly bound to the protein subunits in a number of mating pili, including most recently archaeal conjugation pili (Beltran et al., Nature Communications, 2023). Perhaps this is coincidence, but the suggestion was made by Brinton over 50 years ago that mating pili and filamentous phages were structural homologs!

Lines 190-191) RMSD values are given to a thousandth of an Å. For this to have any significance, the authors must believe that the coordinates of the C-alpha atoms have been determined by both cryo-EM (remarkable) and x-ray fiber diffraction (even more remarkable) to an accuracy of a thousandth of an Å.

Point-by-point response to reviewers

We thank the reviewers for their positive feedback and helpful comments on the manuscript.

Reviewer #1

In the model describing phage assembly and egress depicted in Fig. 1, it is unclear what allows for the inner membrane assembly complex to align with the pIV secretin for efficient egress. The recent Connors et al publication on the pIV (ref 19) proposes the role for the N0 domain of pIV and the Fig. S1 depicts a purple circle and a black line connecting the IM complex to pIV. However, there is no mention of this feature in the figure legend nor in the scheme itself.

Thank you for pointing this out. We have added the additional connection between pIV in the outer membrane and the inner membrane assembly complex pl/pIX in Fig. 1, and have added a description to both figure legends. We have not added the N0 domain to Fig. 6 as the figure becomes too crowded, so an explanation has been added to the legend.

Line 117. The N1 and N2 domains of pIII are not visible and the predictions using alpha fold are poor. What is the predicted disorder score of the linker regions? Would they be vulnerable to proteolysis in the periplasm, and is it possible that due to their negative charge they bind calcium and are compacted in this compartment prior to egress?

We used AlphaFold to predict the structure of the entire pIII protein. The predictions of the N1-N2 domains are actually very good in models 0 and 1. This is shown in Supplementary Fig. 12a, where the N1 and N2 domains are depicted in dark blue, indicating a very high pLDDT score. We used model 0 to make the model of the phage – this has been made clearer in the legend and the methods. It is the C terminal domain and linker region that are less reliably predicted - we have made this clearer by labelling the respective domains in the figure.

We did not need to use the low confidence AlphaFold predictions of the C terminal domain in our model as this could be obtained from our cryoEM data. We did however use the predictions of the linker regions between the C domain and N1-N2 domains to generate the relative positions of the N1-N2 domains in the model of the assembled phage. The fact that the linkers are predicted at very low confidence is not particularly important for our model, because they likely exist in a continuum of different conformations. To address the query on disorder, according to Tunyasuvunakool *et al*, 2021 (<https://doi.org/10.1038/s41586-021-03828-1>), the AlphaFold pLDDT score is a very good prediction of disorder. Values of <50 (which the pIII flexible linker is) should not be interpreted as structures but rather as a prediction of disorder. We have added additional text to the legend and changed the wording in the Results “Model of the f1 filamentous phage” to account for this.

Regarding proteolysis of the N1-N2 domains in the periplasm, this is unlikely to occur. If the receptor binding domains were cleaved, the phage would not be able to infect a new bacterium. It is certainly possible that the linkers may bind something in the periplasm and become more ordered than we see in the egressed and purified form. We have not modified the manuscript in this regard as we do not have any data to support this idea at present.

Line 173. ... termini oriented...

We have changed this.

Reviewer #2

Line 105) “All structures had 5-fold symmetry, with the central filamentous region having additional helical symmetry.” This was quite confusing to me. The outer capsid certainly has helical symmetry in addition to the C5 point-group symmetry. Do they mean here that they only imposed C5 symmetry for an outer capsid map (and did not impose helical symmetry), and did not impose the C5 symmetry on the inner DNA density (but did impose helical symmetry)?

We have made this clearer in the text and pointed readers to the methods section. All final structures had C5 symmetry applied, and the filamentous part had additional helical symmetry applied. This information has also been added to Supplementary table 1. We also ran refinements of the tips without symmetry to attempt to visualise the DNA, and to confirm that the fatty-acid molecules were not symmetry-related artefacts. Further details have been added to the methods section.

Line 108) “resolutions of 2.97 Å, 2.58 Å and 2.81 Å”. Do the authors really believe in the significance of citing resolutions to one-hundredth of an Å given that: a) one can easily change these values by making changes in the mask used; b) the FSC function is rarely monotonic; c) the “gold standard” FSC is not actually measuring resolution, but rather reproducibility.

We agree with the reviewer. We had originally quoted the exact values that are an output from CryoSPARC – which is to 2 decimal places. For consistency with the PDB validation reports, we now quote the PDB values rounded to 1 decimal place.

Lines 120-121) It would be useful if more information were provided about the disulfide bond, rather than just saying that the two were in close enough proximity to form a disulfide. Was density seen for the disulfide? If not, could this be due to the extreme sensitivity of disulfides to electron irradiation (e.g., Pieri et al., PNAS 119, 2022)?

Although density was not observed for a disulphide bond, cysteine residues 354 and 371 were in close enough proximity to form one. The current refinement results in the S atoms being 3.8 Å apart, but alternative rotamers of the cysteine sidechains result in an S-S distance of 2.8 Å, with 3 Å being the cut-off for disulphide bonds in the PDB. As mentioned by the reviewer, we believe the loss of the disulphide bond is due to radiation damage during data collection. We have added additional text and the reference to clarify this point.

Line 156) “lipids are not uncommon in the capsids...”. It might also be mentioned that lipids have been observed tightly bound to the protein subunits in a number of mating pili, including most recently archaeal conjugation pili (Beltran et al., Nature Communications, 2023). Perhaps this is coincidence, but the suggestion was made by Brinton over 50 years ago that mating pili and filamentous phages were structural homologs!

This is interesting and we have added a sentence and reference to the paper mentioned.

Lines 190-191) RMSD values are given to a thousandth of an Å. For this to have any significance, the

authors must believe that the coordinates of the C-alpha atoms have been determined by both cryo-EM (remarkable) and x-ray fiber diffraction (even more remarkable) to an accuracy of a thousandth of an Å.

Again, we reported the values that were a direct output from the software used but we acknowledge the point. We have changed all our RMSD values to round to 1 decimal place.